

# Resistance training with different repetition duration to failure: effect on hypertrophy, strength and muscle activation

Lucas Túlio Lacerda[1,2,3,4], Rodrigo Otávio Marra-Lopes[1], Marcel Bahia Lanza[1,5], Rodrigo César Ribeiro Diniz[1], Fernando Vitor Lima[1], Hugo Cesar Martins-Costa[1,4], Gustavo Ferreira Pedrosa[1,6], André Gustavo Pereira Andrade[1], Armin Kibele[7] and Mauro Heleno Chagas[1]

[1] Weight Training Laboratory, School of Physical Education, Physiotherapy and Occupational Therapy, Universidade Federal de Minas Gerais, Belo Horizonte, Minas Gerais, Brazil
[2] Department of Physical Education and Sports, Centro Federal de Educação Tecnológica, Belo Horizonte, Minas Gerais, Brazil
[3] Universidade do Estado de Minas Gerais, Divinópolis, Minas Gerais, Brazil
[4] Department of Physical Education, Pontifícia Universidade Católica de Minas Gerais, Belo Horizonte, Minas Gerais, Brazil
[5] Department of Physical Therapy and Rehabilitation, School of Medicine, University of Maryland, Maryland, Baltimore, United States of America
[6] Aeronautical Instruction and Adaptation Centre, Brazilian Air Force, Lagoa Santa, Minas Gerais, Brazil
[7] Institute for Sports and Sport Science, University of Kassel, Möncheberg straße, Kassel, Germany

Corresponding author
Mauro Heleno Chagas,
mauroufmg@hotmail.com

## ABSTRACT

**Background**. This study investigated the effects of two 14-week resistance training protocols with different repetition duration (RD) performed to muscle failure (MF) on gains in strength and muscle hypertrophy as well as on normalized electromyographic (EMG) amplitude and force-angle relationships.

**Methods**. The left and right legs of ten untrained males were assigned to either one of the two protocols (2-s or 6-s RD) incorporating unilateral knee extension exercise. Both protocols were performed with 3–4 sets, 50–60% of the one-repetition maximum (1RM), and 3 min rest. Rectus femoris and vastus lateralis cross-sectional areas (CSA), maximal voluntary isometric contraction (MVIC) at 30º and 90º of knee flexion and 1RM performance were assessed before and after the training period. In addition, normalized EMG amplitude-angle and force-angle relationships were assessed in the 6th and 39th experimental sessions.

**Results**. The 6-s RD protocol induced larger gains in MVIC at 30º of knee angle measurement than the 2-s RD protocol. Increases in MVIC at 90º of knee angle, 1RM, rectus femoris and vastus lateralis CSA were not significant between the 2-s and 6-s RD protocols. Moreover, different normalized EMG amplitude-angle and force-angle values were detected between protocols over most of the angles analyzed.

**Conclusion**. Performing longer RD could be a more appropriate strategy to provide greater gains in isometric maximal muscle strength at shortened knee positions. However, similar maximum dynamic strength and muscle hypertrophy gains would be provided by protocols with different RD.

# INTRODUCTION

Repetition duration (RD) is an important feature of a resistance training program (*ACSM, 2009*) influencing the strength gains and hypertrophy (i.e., quadriceps femoris muscles) (*Chaves et al., 2020*; *Tanimoto & Ishii, 2006*). Nevertheless, the systematic effect of the RD on resistance training is not yet fully understood (*Davies et al., 2017*; *González-Badillo et al., 2014*). It has been reported that measurements on isokinetic devices showed poor training and sports specificity (e.g., reduced ecological validity) and the lack of equalization of resistance training protocols would be some of the limitations presented by studies that investigated the influence of RD (*González-Badillo et al., 2014*). Moreover, the absence of registration and/or poor control over the RD, especially during protocols to muscle failure (MF), may hamper its meaning for the effectiveness isoinertial exercises (*González-Badillo et al., 2014*). Hence, RD control and comparability between training protocols must be considered to be mandatory for a proper understanding of the RD effect in a resistance training program.

A meta-analysis on the RD effect on muscle hypertrophy (including only studies with protocols performed to MF) concluded that similar muscle hypertrophy responses may be observed when performing RD between 0.5-s and 8-s (*Schoenfeld, Ogborn & Krieger, 2015*). This result suggests that a wide RD range may be employed in order to produce muscle hypertrophy. However, in addition to RD, the meta-analysis also included studies with variations in the load (*Schuenke et al., 2012*), and studies with training protocols performed until MF or not (*Tanimoto & Ishii, 2006*). Consequently, the results of the meta-analysis cannot be attributed to the manipulation of RD only. Previous studies have suggested that muscle strength and hypertrophy are influenced by the load [(e.g., percentage of one repetition maximum - %1RM)] (*Lasevicius et al., 2018*) and by the RD (*González-Badillo et al., 2014*; *Tanimoto & Ishii, 2006*). Therefore, given that different variables combined may simultaneously influence the chronic responses induced by strength training (*ACSM, 2009*), the effect of RD only within a resistance training to MF while controlled for the load remains unknown.

Another aspect to be considered in studies investigating the impact of RD on muscle hypertrophy relates to the use of different assessment instruments (e.g., biopsy, magnetic resonance imaging or ultrasound) and assessment locations on the muscle (e.g., 50% of the femur length). The cross-sectional area (CSA) is a valid measure of muscle hypertrophy. However, single-point measurements somewhere along the muscle length may not adequately represent the entire muscle hypertrophic response (*Noorkoiv, Nosaka & Blazevich, 2014*). Thus, a CSA analysis including several assessment locations along the muscle length may possibly provide a more accurate depiction of the muscle hypertrophic response in comparison with a specific region along the muscle length (*Noorkoiv, Nosaka & Blazevich, 2014*) and, therefore, a more accurate analysis of hypertrophy gains after resistance training programs as well.

Protocols performed with different RDs enforce different mechanical responses, with higher force values for shorter RD (*Sampson, Donohoe & Groeller, 2014*; *Tanimoto & Ishii, 2006*). As a consequence, different gains in muscle hypertrophy may be induced through resistance training with different RDs (*Gonzalez et al., 2016*). However, *Sampson & Groeller (2016)* showed that a resistance training protocol performed with faster movements (shorter RD) produced similar muscle hypertrophy when compared to a protocol with slower movements (longer RD). Given that the faster movements were not executed with the maximum number of repetitions, the results by *Sampson & Groeller (2016)* remain inconclusive about the RD effect during resistance training to MF. In addition, the protocols with faster movements were performed with a time under tension (TUT, up to three times shorter than the protocol with slower movement) and also higher training volume. The similar muscle hypertrophy observed between protocols reinforce the argument about the impact of mechanical tension (force applied by external resistance to the musculature) to induce adaptations. In this sense, the higher training volume and TUT performed during the longer RD protocol were probably the balance factors in relation to the greater magnitude of mechanical tension observed during protocols with shorter RD (verified by higher peak force values) (*Sampson, Donohoe & Groeller, 2014*), inducing to similar muscle hypertrophy.

In addition, it has been shown that protocols with shorter RD performed to MF presented higher degrees of normalized electromyographic (EMG) amplitude compared to protocols performed with longer RD (*Sakamoto & Sinclair, 2012*). An increase in the EMG amplitude is associated either with a higher motor unit recruitment or an increase in the firing frequency of the motor units (*Hunter, Duchateau & Enoka, 2004*). Both factors would contribute to chronic adaptations associated with resistance training (*Schoenfeld et al., 2014*). Therefore, considering that the magnitude of the mechanical tension and EMG amplitude would be determinant factors of neuromuscular adaptations (*Gehlert et al., 2015*), protocols with shorter RD performed to MF (consequently higher number of repetitions) should theoretically provide superior responses of muscle hypertrophy when compared to protocols performed with longer RD.

As above-mentioned, protocols incorporating different RDs and repetition numbers provide different mechanical (*Sampson, Donohoe & Groeller, 2014*) and neurophysiological responses (*Lacerda et al., 2016*; *Sampson, Donohoe & Groeller, 2014*). As reported for muscle hypertrophy, it has been presented that these factors can influence muscle strength throughout resistance training. In this sense, the review by *Davies et al. (2017)* verified only a trend for larger gains in muscular strength (measured by 1RM performance) for protocols with shorter RDs and moderate intensities (60–79% 1RM) compared to longer RDs. Unfortunately, protocols with exercises leading to MF were not considered in this review. Moreover, the 1RM test does not provide information on maximum force values in different joint angles. In particular, the 1RM test fails to provide information about maximal force values in specific sections of the ROM where a mechanical disadvantage may possibly occur to explain the different adaptations to RT (*Van den Tillaar, Saeterbakken & Ettema, 2012*). As a consequence, maximum voluntary isometric contractions (MVIC) should

be analyzed across a range of different joint angles to properly understand the effects of different RDs (*Alegre et al., 2014*).

In the past, studies showed that different RDs evolved to different force–angle relationships across the ROM. This was particularly true for the beginning and the end of the muscle actions (*Sampson, Donohoe & Groeller, 2014*; *Tanimoto & Ishii, 2006*). Protocols with shorter RDs require faster movements. Therefore, they lead to larger peak forces at the beginning of the concentric action (e.g., lengthened position during knee extension) compared to protocols with longer RDs (*Sampson, Donohoe & Groeller, 2014*). At the end of the ROM (e.g., shortened position during knee extension), a decrease in force is observed when faster movements are performed. In contrast, protocols with longer RDs come along with less variation in the force response throughout the ROM, while larger force values appear at the end of the concentric actions (*Sampson, Donohoe & Groeller, 2014*; *Tanimoto & Ishii, 2006*). All in all, varied strategies to apply force throughout the concentric action incorporating different RDs may promote different increases in maximal isometric strength at specific points across the ROM. As a consequence, it was the aim of this study to compare the effects of two protocols with different RDs performed to MF on measures of maximal strength (1RM and MVIC) and muscle hypertrophy (CSA). A secondary aim was to compare the effects of these RD strategies on features of the normalized EMG amplitude-angle and force–angle relationships during both protocols execution. Based on our previous arguments, we hypothesized that larger increases in the 1RM and the CSA would be induced by a protocol with shorter RDs. In addition, the MVIC gains were expected to be different in specific areas across the ROM. In particular, larger forces were expected for faster training protocol at 90° of knee flexion (stretched position) and for slower protocol at 30° of knee flexion (shortened position).

## MATERIALS & METHODS

### Study design

In the present study, a repeated measures design was adopted. Given that the unilateral exercise model reduces inter-subject variability, it can serve to increase statistical power, as well as reduce the time and cost of a study (*MacInnis et al., 2017*). Volunteers performed two resistance training protocols with two RDs (2-s or 6-s RD protocol) for 14 weeks. The left and the right legs were randomly assigned and balanced for limb dominance to either one of the protocols. Pre and post-test measures included: CSA, MVIC and 1RM tests. To assess the lower limb dominance voluntaries were asked to answer the following question: "If you would shoot a ball on a target, which leg would you use to shoot the ball?"

In session 1, limb dominance was determined, volunteers were familiarized with all the procedures, and training protocols were assigned to each limb. In the next session, ultrasound images were recorded to determine rectus femoris and vastus lateralis CSA. The strength tests (MVIC and 1RM) were conducted in sessions 3 and 4 separated by at least 48 h. Next, subjects trained from sessions 5 to 39 for a total of 14 weeks and five training sessions per week. The training sessions were separated by at least 24 h. For each week, subjects trained their left or right either on days 1, 3, and 5, or on days 2 and 4 in an

alternating way. Through this training schedule, a minimum of 48 h inter session rest was provided for each leg. In sessions 6 and 39 (for each protocol), the rectus femoris and vastus lateralis EMG amplitude were assessed through surface EMG while participants performed their respective training protocols. In session 40, separated between 72 and 120 h from the last training session, the post-test ultrasound measurements were conducted similar to session 2. Finally, in session 41, the MVIC and 1RM post-tests were executed for both lower limbs.

## Participants and ethics

The sample size calculation was performed by using the software G.Power for Windows version 3.1.9.7 (Düsseldorf, Germany) and by following the guidelines proposed by *Beck (2013)*, with a priori statistical power (1 - ß) = 0.80, effect size (f) = 0.57 and 5% significance level. Number of groups = 2 (protocols); number of measures = 2 (pre-post); correlation between measures = 0.90; sphericity = 1. For the sample size calculation, we used the absolute AST values of the vastus lateralis muscle from a previous study from our own laboratory, which was carried out with an experimental design similar to the present study (*Lacerda et al., 2020*).Ten males aged between 18 and 30 years (mean ± SD: age = 23.1 ± 4.63 years; body height = 1.72 ± 0.07 m; body mass = 68.4 ± 9.46 kg; body fat percentage = 14.03 ± 6.56%) participated in this study.

The inclusion criteria for participation were: (1) no resistance training during the last six months; (2) no functional limitations that could influence the 1RM test or the training protocols; and (3) no use of pharmacological substances or ergogenics supplements, and no other modes of resistance exercise during the study period. Subjects were informed about the study aims, procedures, and risks prior to signing an informed consent form. The ethics committee of the Federal University of Minas Gerais approved this study (approval number: 79108117.5.0000.5149), which complied with the Declaration of Helsinki. Additionally, each subject was instructed not to engage in any physical activity immediately before the testing sessions and to maintain the same diet before each session.

## Experimental Session 1(anthropometric measurements)

After receiving information about the goals and the purpose of the study and giving written consent, the volunteers answered the Physical Activity Readiness Questionnaire (PAR-Q). Next, they were submitted to an anamnesis examining possible limitations related to the study participation. In addition, body height, mass, and fat percentage (skinfold thickness) measurements were conducted. As a next step, volunteers were positioned on a seated knee extension machine (Master, Minas Gerais, Brazil) while maintaining a hip angle of 110° (angle between the backrest and the equipment seat). For measurement purposes, the lateral epicondyle of the femur was aligned with the rotational axis of the device and the pad of the device placed approximately three cm above the medial malleolus. These positions were registered for future replication in the subsequent tests and training sessions. All test sessions were held at the same time of the day for each volunteer.

## Experimental sessions 2 and 40 (CSA - ultrasound measurements)

During these sessions, ultrasound images were recorded for the CSA analysis of the rectus femoris and vastus lateralis muscles. The acquisition procedure for the CSA images was conducted as described by *Noorkoiv, Nosaka & Blazevich (2010)* and *Lacerda et al. (2020)*. Initially, volunteers remained in a dorsal decubitus position on a stretcher for 15 min. During this period, the anterior regions of both thighs were marked to identify the reference points for the ultrasound image acquisition. Next, the major trochanters and lateral epicondyles of the femurs were identified, and femur length was measured (Fig. 1A). From the proximal end of thigh, 40, 50, 60, and 70% of femur length were identified and marked on volunteer's skin by using a tape measure and a pachymeter positioned parallel to the intercondylar line. Then, a line with a microporous adhesive tape was attached two cm from each percentage point on the thigh (Fig. 1B) to delimitate the probe guide area for the ultrasound image acquisition (Fig. 1C). Finally, the distances between the intercondylar line and each percentage point on the thighs were recorded for post-test replication. The procedures used to acquire images in the pre-test were the same for the post-test session (40th session). The latter was started no earlier than 72–120 h following the last training session.

An ultrasound device (MindRay DC-7, Shenzhen, China) was used in an extended-field-of-view mode with a four cm linear transducer. The equipment was configured with 10 MHz frequency, an acquisition rate of 21 frames/s, a depth for the image capture ranging from 7.7 and 9.7 cm, and a gain between 50 and 64 dB. The settings were adjusted for each subject to produce the clearest images of the analyzed muscles. The same experienced examiner ($\sim$120 h of training and 600 images acquired before of the study) conducted the acquisition of two images for each of the given femur percentage lengths (40, 50, 60, and 70%). For the acquisition procedure, the probe was placed transversely in parallel to the intercondylar line using a coupled guide on the subject's thigh (probe guide) (Fig. 1C). This procedure was performed with constant speed (controlled by a metronome) and lasted between 12 and 15 s depending on the subject's thigh circumference. Sixteen images per subject were obtained for the rectus femoris and vastus lateralis CSA analysis (8 pre-test + 8 post-test). Following the acquisition procedure, CSAs of each muscle scan were manually demarcated by a blinded examiner using specific software (OsiriX MD 6.0, Bernex, Switzerland) (Fig. 2). For the data analysis, the rectus femoris and vastus lateralis CSA mean values were calculated using two images acquired at each percentage of the femur length. Finally, based on the 40, 50, 60, and 70% length measurements, the sum of four CSAs of each analyzed muscle was calculated, generating a summary CSA value per muscle to avoid a possible misinterpretation based on one measurement site only (*Noorkoiv, Nosaka & Blazevich, 2010*). This value was used in the statistical data analysis.

## Experimental sessions 3, 4 and 41 (strength tests)

The strength tests were conducted during the third session in order to familiarize the subjects with the procedures to be performed during the following session. After positioning the participants in the equipment, a familiarization MVIC test was conducted encompassing two attempts of 5 s in duration with knee flexion angles of 30° and 90° (knee extended =

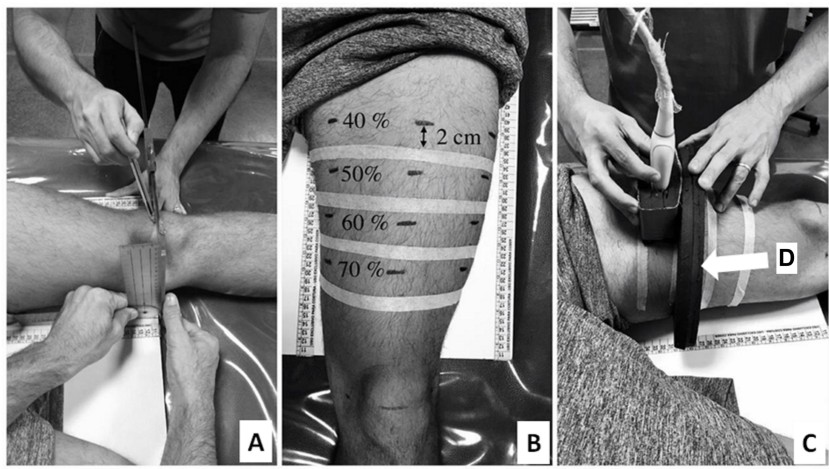

**Figure 1** Thigh marking procedures (A and B) and ultrasound images acquisition (C). Probe guide (indicated by white arrow) (D).

0°). MVIC tests were conducted with both legs with 2-minute rest periods between each attempt (*Lacerda et al., 2020*). The testing order was randomized between legs. The same order was maintained during the post-test session. The highest force value registered for each attempt at knee flexion angles of 30° and 90° was used in further data analyses. During the MVIC test, a verbal command was given on which the subject exerted a maximum force against the fixed lever of the knee extensor machine. Visual feedback of the force trace was provided to the subject as well as verbal instruction from the examiners to achieve maximum strength. The load cell raw data (Tedea, Bavaria, Germany) were converted into digital data (Biovision, Wehrheim, Germany) and filtered through a 4th-order Butterworth low-pass filter with a cutoff frequency of 10 Hz.

The 1RM test familiarization was performed 10 min after the completion of the MVIC test. Initially, according to procedures described by *Lacerda et al. (2016)* and *Lacerda et al. (2020)*, subjects performed 10 repetitions without any weight on the equipment. The 1RM was determined in concentric mode within a maximum of 6 attempts with 5-minutes rest periods in between (*Lacerda et al., 2016*; *Lacerda et al., 2020*). In addition, a 5-minute rest period was given between the tests conducted with each of the lower limbs.

In session 4, the MVIC and 1RM tests of the familiarization session were repeated. These tests were also repeated in the 41supst experimental session with a rest interval of at least 48 h following the previous session 40 (ultrasound measurements). The data measured in sessions 4 and 41 were used for statistical analysis.

## Experimental sessions 5 to 39 (training period)

After the initial testing period, the 14-week training commenced (35 training sessions). All participants completed 100% of the training sessions. The experimental protocols consisted of 3–4 sets at 50–60% 1RM with 3-minute rest periods in between. In the 2-s RD protocol, subjects completed each repetition in 2 s (1 s concentric, 1 s eccentric). In the 6-s RD protocol, subjects perform each repetition in 6 s (3 s concentric, 3 s
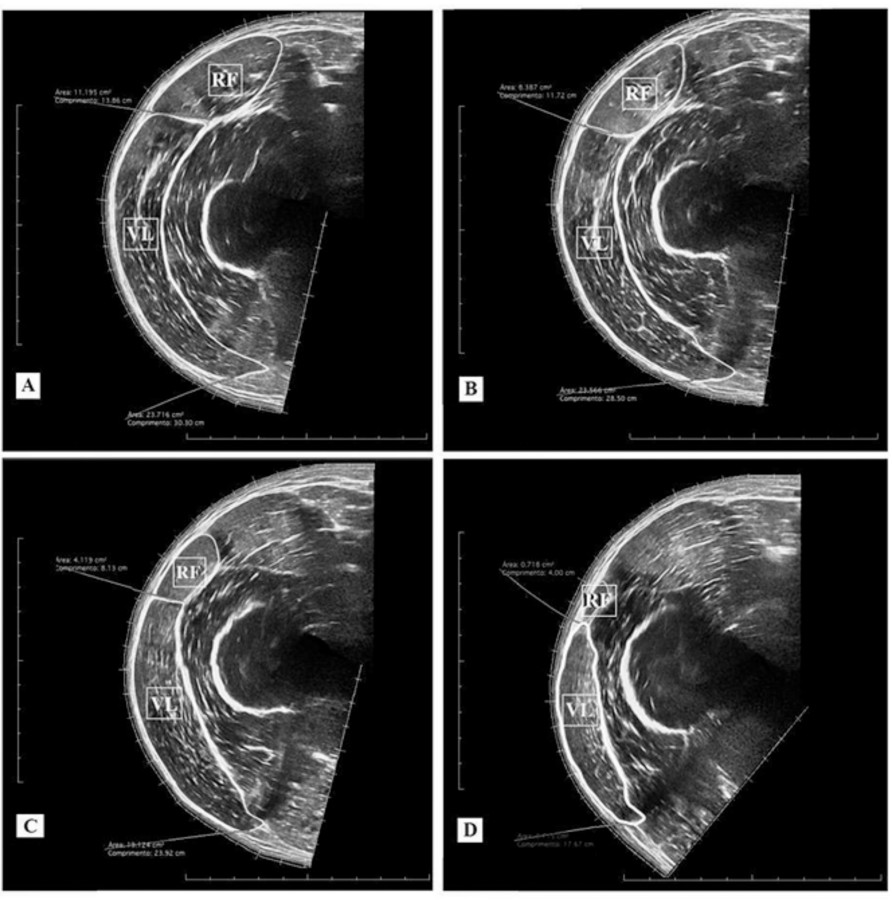

**Figure 2** Ultrasound images and cross-sectional areas (CSA) at 40% (A); 50% (B), 60% (C), and 70% (D) of femur length. Rectus femoris (RF) and vastus lateralis (VL).

eccentric). The protocols complied with recommendations for resistance training and muscle hypertrophy (*ACSM, 2009*). Previously, training protocols with similar concentric and eccentric durations were already investigated in our laboratory or in others' (*Lacerda et al., 2016*; *Lacerda et al., 2020*; *Sakamoto & Sinclair, 2012*). For both protocols, all sets were executed until the subjects were unable to complete the concentric action within the required ROM (70°).

During the first two weeks, training sessions included 3 sets at 50% of 1RM. At third week (6[th] training session), the load was increased to 60% of 1RM. From week 9 (20[th] training session) until the end of the training period, one more set was added so the participants performed 4 sets at 60% of 1 RM. Given that any variation of the load characteristics in addition to the RD could possibly bias the training adaptations, the present study controlled the load configuration and progression.

Every two weeks, beginning in the third week (6[th] training session), 1RM tests for both legs were re-assessed on a weekly basis before the first training session. A 10-minute rest period between the 1RM test and the start of the training session was provided. During

these sessions, the 1RM test was conducted at the same day time as in the pre-test to standardize the circadian rhythm, which may possibly influence strength performance.

## Experimental sessions 6 and 39 (2nd and 35th training sessions) (force and electromyography measurements)

The surface EMG procedure (Biovision, Wehrheim, Germany) followed the recommendations by *Hermens et al. (2000)*. For the rectus femoris and vastus lateralis muscles, bipolar surface electrodes (Ag/AgCl - 3M-2223, Brazil) were aligned parallel to the muscle fiber orientation. Prior to the electrode placement above the muscle bellies, the skin areas were shaved, cleaned with alcohol using a cotton pad. The inter-electrode distance was 4 cm which each electrode to be placed 2 cm distant from the center of the muscle belly. The ground electrode was attached above the patella. After the electrode attachment, a silk paper was used to assess their positions as well as the patella and other relevant points on the skin. In addition, the subject's two thighs were photographed with the electrodes positioned. These procedures were conducted in the 6th session to map the electrode positions on the thigh and to verify high reproducibility in the post-test measurements (39th session).

To measure the ROM and the muscle action durations during both protocols, the angular displacement was recorded using a potentiometer (aligned with volunteer's knee-joint). For all training sessions, this device was coupled to the rotational axis of the knee extension device. The potentiometer raw data were converted into angular displacement data and filtered through a 4th-order Butterworth low-pass filter with a cutoff frequency of 10 Hz. The duration of each muscle action was comprised of the time between the maximum (100° of knee flexion) and minimum (30° of knee flexion) angular positions. Thus, the duration of the concentric action corresponded to the period between the maximum and minimum angular positions. In turn, the duration of the eccentric action corresponded to the time between the minimum and maximum angular positions. Additionally, concentric/eccentric durations and the RDs were determined throughout the angular displacement time. This potentiometer data provided online information on a laptop screen for the subjects regarding the duration and ROM data of each muscle action throughout the training sessions and tests (*Lacerda et al., 2016*; *Lacerda et al., 2020*). Moreover, a metronome was used to help subjects maintain pre-established RDs.

All electromyographic, load cell, and potentiometer signals were synchronized and converted by an A/D board (Biovision, Wehrheim, Germany) with a sampling rate of 4,000 Hz. DasyLab software (Version. 11.0; Measurement Computing Corporation, Massachusetts, USA) was used to record and process the data. The methodological procedures to record force measurements were detailed in the strength tests section. The electromyographic data acquisition was amplified (factor 500) and filtered (4th-order Butterworth band-pass filter of 20–500 Hz) to calculate the EMG amplitude as the root mean square. Before commencing each experimental session (6th or 39th), subjects were asked to perform a MVIC test for 5 s on the knee extension machine exercise at 60° knee flexion (controlled by the potentiometer). The highest force and EMG amplitude values in the MVIC test were used as a reference for the normalization of the subsequent measurements in the exercise protocols. The EMG amplitude during the MVIC was

measured over a 1 s period from 500 ms before the MVIC peak force to 500 ms after (*Piitulainen et al., 2013*). The mean force and EMG amplitude of the concentric muscle actions for each 10° knee flexion area (100°–90°, 90°–80°, up to 40–30°) was calculated and normalized by the reference values from the normalization test. As a result, relatives force and EMG amplitude × knee-joint angle curves (normalized force and EMG-angle) were assessed. This procedure was performed for each protocol. For the acquisition of the force and EMG amplitude values during experimental sessions 6 and 39, participants performed 3 sets with 50% of the previous 1RM value in each protocol.

## Statistical analyses

Statistical analysis was performed with SPSS for Windows version 20.0 (SPSS, Inc., Illinois, USA). The normal distribution was verified by the Shapiro–Wilk test. All data were expressed as mean ± SD. For the estimation of effect sizes, eta squared ($\eta^2$) values are considered to reflect the magnitude of the differences (effect size) in each treatment with values $\leq 0.010$ expressing a trivial effect; values between 0.010 and 0.059 expressing a small effect; values between 0.060 and 0.139 a moderate effect, and values $\geq 0.140$ a large effect.

Initially, paired sample $t$-tests were used to test for differences between the training protocols in baseline values for the main variables analyzed (CSA, 1RM and MVIC), as well as for EMG amplitude and force values obtained during the normalization. The differences between the training protocols in the CSA scores were analyzed through a two-way repeated-measures analysis of variance test (ANOVA) for each muscle separately, having protocol (2-s RD or 6-s RD) and time (pre and post-test) as factors. In case of significant F-values a Bonferroni adjustment was used for comparison purposes. The intra-rater reliability was verified by the intraclass correlation coefficient ($ICC_{[3,1]}$). For the ICC calculations were conducted for both CSA measures (rectus femoris and vastus lateralis) and for both the test sessions (pre and post-test).

Similar to CSA measurements, a two-way repeated-measures ANOVA test (protocol × time) was applied for MVIC at 30° of knee flexion scores. However, the 1RM and MVIC at 90° of knee flexion values were significantly different at baseline. Therefore, the baseline values were considered as a covariate, and an analysis of covariance (ANCOVA) was implemented using a within-subject factors model. In addition, the ICC intersession values for 1RM and MVIC were obtained from measures during the third (familiarization) and fourth (pre-test) sessions. The familiarization and pre-test sessions were separated by at least 48 h.

Normalized EMG amplitude-angle relationships for the rectus femoris and the vastus lateralis muscles were established during the 6[th] and 39[th] sessions to compare EMG amplitude differences between the 2-s and the 6-s RD protocols. A three-way repeated measures ANOVA test (session × protocol × knee joint angle) was conducted to analyze the training effects in the normalized EMG amplitude for each muscle. Similar to EMG amplitude responses, a three-way repeated measures ANOVA test (session × protocol × knee joint angle) was used to compare normalized force–angle relationships in the 6[th] and 39[th] sessions. When necessary, a post hoc Bonferroni honest significant difference test was used to identify the differences reported in the ANOVA's.

Furthermore, the EMG amplitude and force values for each protocol obtained during the normalization test from experimental sessions 6 and 39 were compared by $t$-test. This procedure aimed to identify possible differences in measurements in both lower limbs of the same individual. Thus, the feasibility of comparing the EMG amplitude and force responses of the two training protocols should be established.

In addition, paired sample t-tests were used to compare the RDs, ROM and TUT mean values for all sets during training sessions between investigated protocols. Finally, given the number of repetitions for each protocol does not meet the precepts for a parametric analysis, Wilcoxon test was used to compare the values in this variable for both protocols. This data is presented as median (number repetitions per set) and interquartile interval values. The level of the error probability/statistical significance was set at $p \leq 0.05$ for all statistical tests.

## RESULTS

### CSA

The intra-rater reliability values found in these sessions were 0.99 for both analyzed muscles. No significant interaction was observed between protocol and time for the rectus femoris CSA ($F_{1,9} = 1.889$; $p = 0.203$; $\eta^2 < 0.010$ "trivial"). Also, no significant differences between groups were found (main protocol effect; $F_{1,9} = 0.001$; $p = 0.972$; $\eta^2 < 0.010$ "trivial"). Both training protocols showed significant increases in rectus femoris CSA after training period (main time effect; $F_{1,9} = 64.353$; $p < 0.001$; $\eta^2 = 0.821$ "large") (Fig. 3A).

In addition, no significant interaction was observed between protocol and time for the vastus lateralis CSA ($F_{1,9} = 0.867$; $p < 0.376$; $\eta^2 < 0.010$ "trivial"). No significant difference between the 2-s and the 6-s RD protocols were found (main protocol effect; $F_{1,9} = 0.009$; $p = 0.928$; $\eta^2 < 0.010$ "trivial"). Both training protocols showed significant increases in vastus lateralis CSA after training period (main time effect; $F_{1,9} = 50.664$; $p < 0.001$; $\eta^2 = 0.814$ "large") (Fig. 3B).

### 1RM

The ICC intersession value for 1RM tests was 0.98. There were significant difference in 1RM values at baseline (Pre; $t_9 = 2.688$; $p = 0.025$). When baseline differences in 1RM values were taken into account (within-subject factors ANCOVA), no significant main protocol effect was found ($F_{1,9} = 0.412$; $p = 0.820$; $\eta^2 < 0.010$ "trivial") (Fig. 4).

### MVIC at 30° of knee flexion

The ICC intersession value for MVIC test at 30° of knee flexion was 0.96. A significant interaction was observed between protocol and time for MVIC test at 30° of knee flexion measurements ($F_{1,9} = 6.576$; $p = 0.030$; $\eta^2 = 0.049$ "small"), with higher values for the 6-s RD protocol. Furthermore, a significant main effect of time ($F_{1,9} = 7.581$; $p = 0.023$; $\eta^2 = 0,273$ "large") was detected. Conversely, no significant main effect of protocol was found ($F_{1,9} = 1.990$; $p = 0.192$; $\eta^2 = 0.101$ "moderate") (Fig. 5).

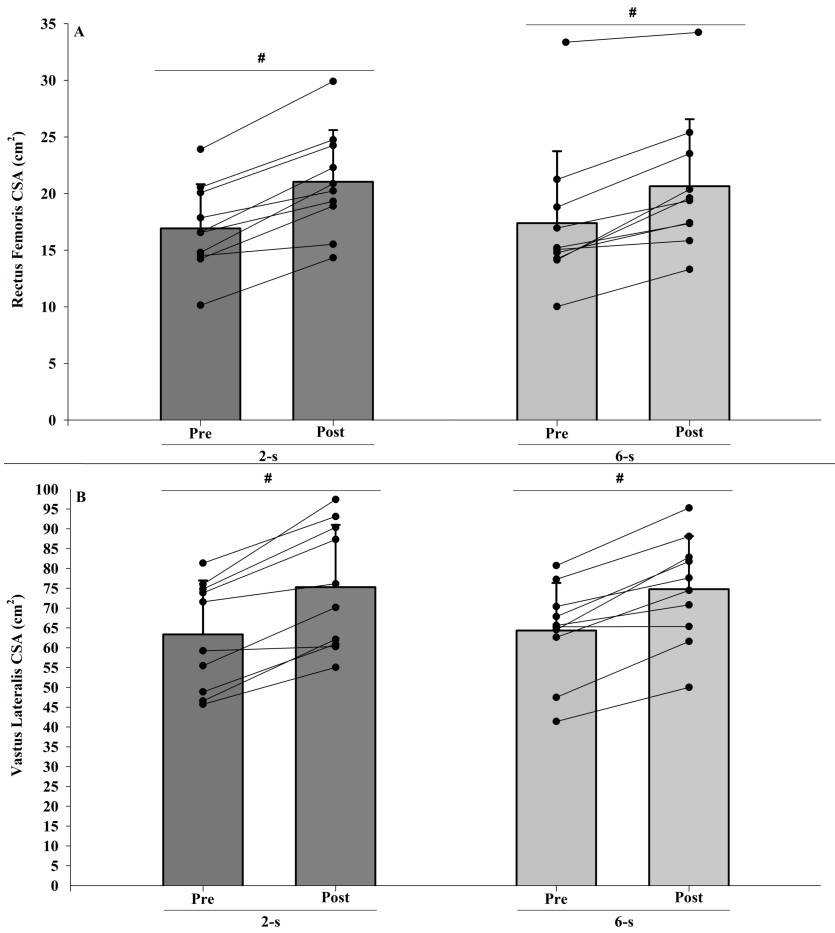

**Figure 3** **Rectus femoris (A) and vastus lateralis (B) cross-sectional areas (CSA) at pre-test to post-test for each training protocol.** Mean (vertical bars); standard deviation (vertical lines); individual values for each training protocol (black circles). # post-test higher than pre-test for both protocols (time main effect).

## MVIC at 90° of knee flexion

The ICC intersession value for MVIC test at 90° of knee flexion was 0.94. There were significant differences at baseline (Pre; $t_9 = 2.640$; $p = 0.027$). When baseline differences in MVIC test at 90° of knee flexion values were taken into account (within-subject factors ANCOVA), no significant main protocol effect was found ($F_{1,9} < 0.001$; $p = 0.996$; $\eta^2 < 0.010$ "trivial") (Fig. 6).

## Normalized EMG amplitude-angle relationship

The EMG amplitude values obtained during the normalization tests showed no significant difference between the 2-s RD and the 6-s RD training protocols (rectus femoris - $t_{19} = 0.701$, $p = 0.503$) (vastus lateralis - $t_{19} = 0.773$, $p = 0.455$).

No significant interaction between time × protocol × knee joint angle was observed for the normalized EMG amplitude data of rectus femoris ($F_{6,54} = 2.025$; $p = 0.143$; $\eta^2 < 0.010$ "trivial") and vastus lateralis muscles ($F_{6,54} = 2.640$; $p = 0.106$; $\eta^2 < 0.010$ "trivial").

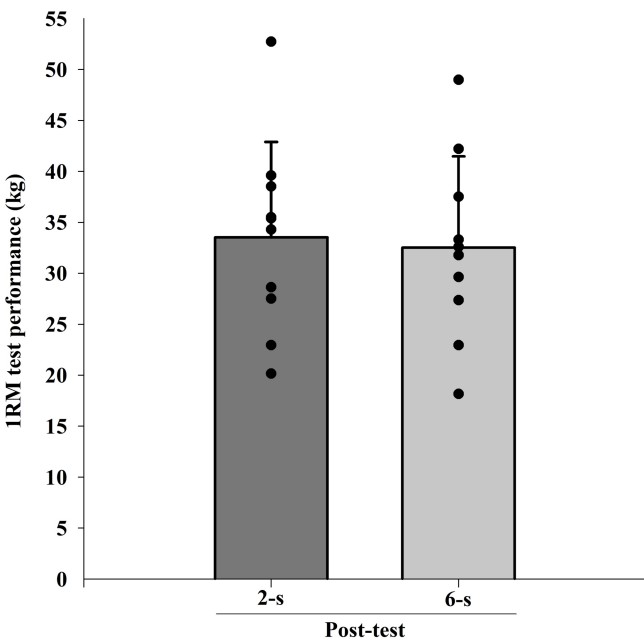

**Figure 4 Maximal dynamic strength (1RM) test at post-test for each training protocol.** Mean (vertical bars); standard deviation (vertical lines); individual values for each training protocol (black circles).

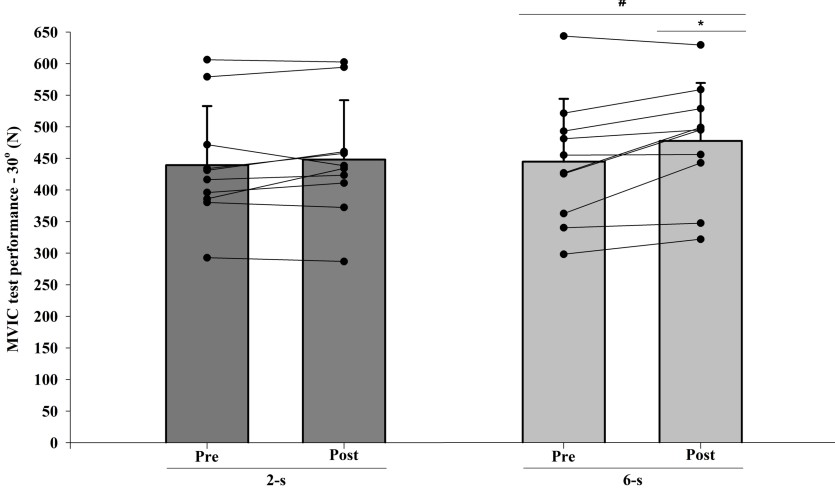

**Figure 5 Maximal isometric strength (MVIC) at 30° of knee-joint angle at pre-test to post-test for each training protocol.** Mean (vertical bars); standard deviation (vertical lines); individual values for each training protocol (black circles). * 6-s RD protocol higher than 2-s RD protocol. # post-test higher than pre-test for both protocols (protocol and time interaction effect).

However, significant differences in the rectus femoris EMG amplitude were observed between the 2-s RD and the 6-s RD training protocols in all knee-joint angles analyzed during $6^{th}$ and $39^{th}$ sessions (protocol × knee-joint angle interaction - $F_{6,54} = 66.554$; $p < 0.001$; $\eta^2 = 0.199$ "large") (Figs. 7A, 7B). The 2-s RD protocol resulted in larger

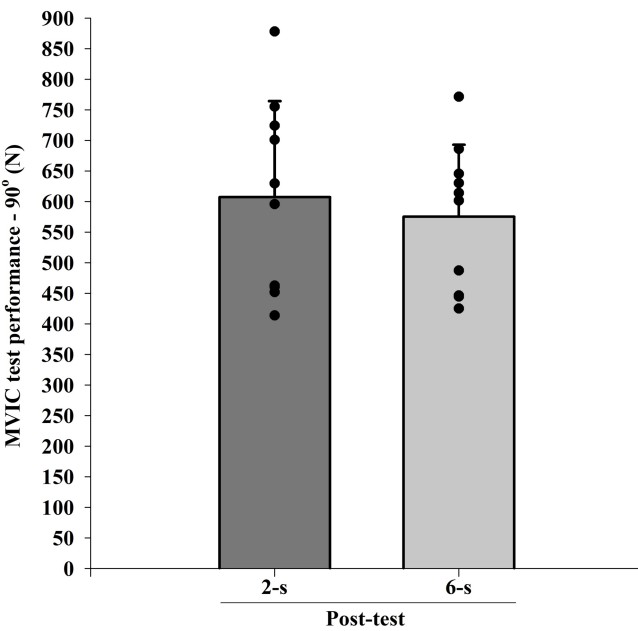

**Figure 6 Maximal isometric strength (MVIC) at 90° (B) of knee-joint angle at post-test for each training protocol.** Mean (vertical bars); standard errors (vertical lines); individual values for each training protocol (black circles).

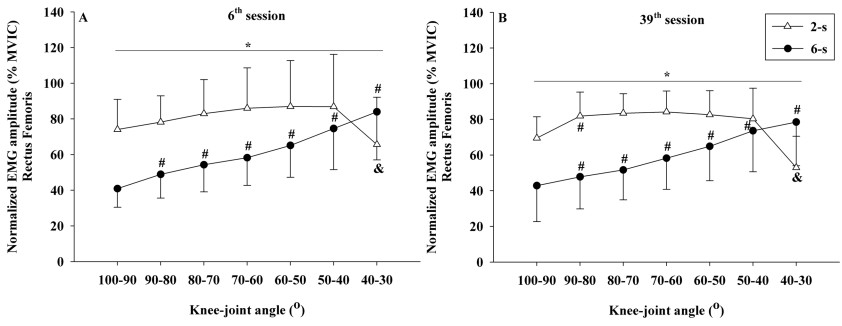

**Figure 7 Rectus femoris concentric normalized EMG amplitude × knee-joint angle curves during 6th (A) and 39th (B) experimental sessions at 2-s and 6-s RD protocols.** Mean values for 2-s RD (white triangles); Mean values for 6-s RD (black circles); standard errors (vertical lines). * Significant difference between protocols. # Higher than previous joint angle (6-s RD protocol). & Lower than all previous joint angles, except for 100–90° (2-s RD protocol).

normalized EMG amplitude scores in six of the seven knee-joint angles analyzed (100−90° to 50−40°). Conversely, the 6-s RD protocol provided significantly larger rectus femoris EMG amplitude in the last knee-joint angle (40–30°) only. The same results were verified for the vastus lateralis EMG amplitude (protocol × knee joint angle interaction - $F_{6,54} = 51.007$; $p < 0.001$; $\eta^2 = 0.179$ "large") (Figs. 8A, 8B).

In addition, no significant interactions were detected between time × knee joint angle for rectus femoris ($F_{6,54} = 2.643$; $p = 0.070$; $\eta^2 < 0.010$ "trivial") and vastus lateralis

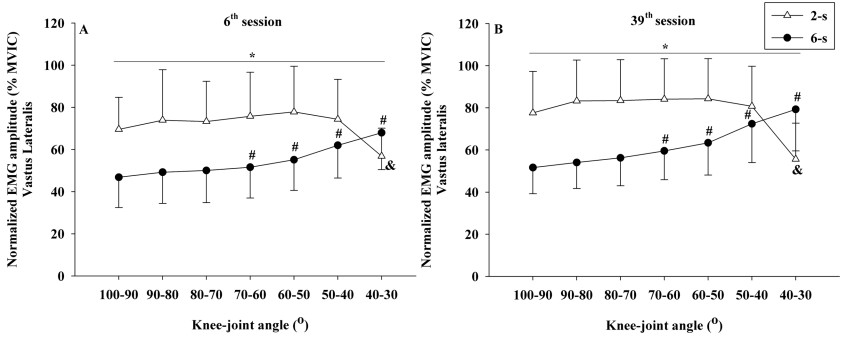

**Figure 8  Vastus lateralis concentric normalized EMG amplitude ×knee-joint angle curves during 6th (A) and 39th (B) experimental sessions at 2-s and 6-s RD protocols.** Mean values for 2-s RD (white triangles); Mean values for 6-s RD (black circles); standard errors (vertical lines). ⋆ Significant difference between protocols. # Higher than previous joint angle (6-s RD protocol). & Lower than all previous joint angles (2-s RD protocol).

muscles ($F_{6,54} = 0.414$; $p = 0.866$; $\eta^2 < 0.010$ "trivial"). Also, no significant interactions were detected between time × protocol for rectus femoris ($F_{1,9} = 0.168$; $p = 0.692$; $\eta^2 < 0.001$ "trivial") and vastus lateralis muscles ($F_{1,9} = 0.014$; $p = 0.910$; $\eta^2 < 0.001$ "trivial"). No significant main effect for the time factor (rectus femoris: $F_{1,9} = 0.143$; $p = 0.714$; $\eta^2 < 0.010$ "trivial") (vastus lateralis: $F_{1,9} = 2.434$; $p = 0.153$; $\eta^2 = 0.043$ "small") was detected. In contrast, significant main effects were found for the training protocol (rectus femoris: $F_{1,9} = 29.46$; $p < 0.001$; $\eta^2 = 0.202$ "large") (vastus lateralis: $F_{1,9} = 16.131$; $p = 0.003$; $\eta^2 = 0.225$ "large") and for knee-joint angle (rectus femoris: $F_{6,54} = 15.673$; $p < 0.001$; $\eta^2 = 0.111$ "large") (vastus lateralis: $F_{6,54} = 10.179$; $p > 0.001$; $\eta^2 = 0.037$ "small").

## Normalized force–angle relationship

The force values obtained during the normalization showed no significant difference between both training protocols analyzed ($t_{19} = 0.732$; $p = 0.478$).

Significant differences in the normalized force–angle relationship were observed between the 2-s RD and the 6-s RD training protocols during 6th and 39th sessions (protocol × knee-joint angle × time - $F_{6,54} = 2.652$; $p = 0.025$; $\eta^2 = 0.033$ "small"). In the 6th session, the 2-s RD protocol exhibited significantly larger normalized force values in the first four knee-joint angles analyzed ($100-90°$ to $70-60°$). The same was true for the first three knee-joint angles ($100-90°$ to $80-70°$) in the 39th experimental session. In contrast, for the 6-s RD protocol larger normalized force values were found in the last two knee-joint angles ($50-40°$ $40-30°$) during sessions 6 and 39. In addition, significant changes in the force ×angle relationships between training sessions were detected only for the 2-s RD protocol. These changes were related to a reduction in the force values from $70-60°$ until $40-30°$ of knee flexion (Figs. 9A, 9B).

Moreover, significant interactions were found between the time x knee joint angle ($F_{6,54} = 14.243$; $p < 0.001$; $\eta^2 = 0.011$ "small") and the protocol x knee-joint angle ($F_{1,9} = 296.591$; $p < 0.001$; $\eta^2 = 0.370$ "large"). No significant interaction was observed

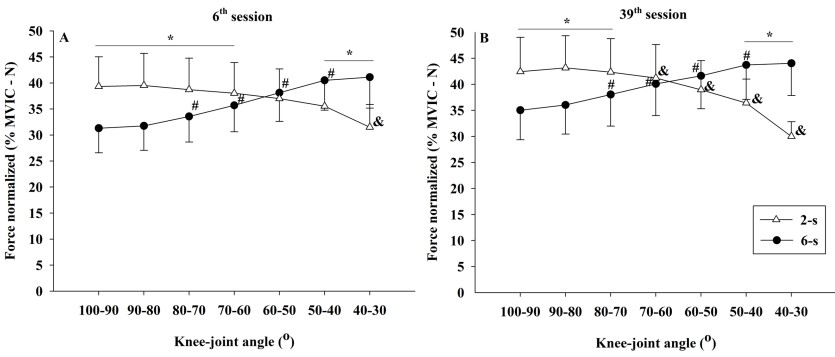

**Figure 9** Concentric normalized force × knee-joint angle curves during 6$^{th}$ (A) and 39$^{th}$ (B) experimental sessions at 2-s and 6-s RD protocols. Mean values for 2-s RD (white triangles); Mean values for 6-s RD (black circles); standard errors (vertical lines). ⋆ Significant difference between protocols. # Higher than previous joint angle (6-s RD protocol). & Lower than previous joint angle (2-s RD protocol).

for the interaction of time × protocol ($F_{1,9} = 2.483$; $p = 0.150$; $\eta^2 < 0.010$ "trivial"). Finally, no significant main effect for time ($F_{1,9} = 1.807$; $p = 0.212$; $\eta^2 = 0.074$ "moderate") and for protocol ($F_{1,9} = 1.555$; $p = 0.703$; $\eta^2 < 0.010$ "trivial"), but a significant main effect for knee-joint angle ($F_{6,54} = 7.453$; $p < 0.001$; $\eta^2 = 0.026$ "small") were identified.

### RD, TUT, Number of repetitions and ROM

As expected, 6-s RD protocol showed longer average RD than 2-s RD protocol (2.04 ± 0.08 s; 5.98 ± 0.09 s, respectively; $t_{69} = 284.488$, $p < 0.001$). In addition, larger TUT mean values were observed in the 6-s RD protocol (mean for all sets = 43.47 ± 10.92; 1$^{st}$ set = 52.5 ±15.83, 2$^{nd}$ set = 42.58 ± 12.25, last set (3$^{rd}$ or 4$^{th}$) = 36.98 ± 10.89) as compared to the 2-s RD protocol (mean for all sets = 30.51 ± 7.52; 1$^{st}$ set = 38.1 ±10.55, 2$^{nd}$ set = 29.57 ± 7.98, last set (3$^{rd}$ or 4$^{th}$) = 25.09 ± 7.17) ($t_{69} = 15.951$; $p < 0.001$). In regard to the number of repetitions, the Wilcoxon test showed significantly larger median values for the 2-s RD protocol (median for all sets = 14 [12–17]; 1$^{st}$ set = 18 [21.25–17], 2$^{nd}$ set = 14.5 [16.25–12.75], last set (3$^{rd}$ or 4$^{th}$) = 12 [14–11]) as compared to the 6-s RD protocol (median for all sets = 7[6–8]; 1$^{st}$ set = 9 [8–10], 2$^{nd}$ set = 7 [6–8], last set (3$^{rd}$ or 4$^{th}$) = 6 [5–7]) ($U_{69} = 7.294$; $p < 0.001$). Last not least, but no significant differences were detected between the 2-s RD and the 6-s RD protocols in the ROM average values (2-s RD: 70.77 ± 0.79° -; 6-s RD: 70.99 ± 0.65° -; $t_{69} = 1.903$, $p = 0.06$).

## DISCUSSION

The purpose of this study was to compare the strength and muscle hypertrophy responses induced by two protocols with different RDs performed to MF. In addition, we aimed to verify the effects of these RD strategies on knee extension force and EMG amplitude in the rectus femoris and vastus lateralis muscles. To the best of our knowledge, no other studies have compared these chronic adaptations for resistance training with different RD to MF by matching load set, and rest and, in addition to that, using ultrasound measurements from four different parts of the muscle. The main results of the present study were: (1)

the 2-s RD protocol showed similar effects in maximum dynamic strength and muscle hypertrophy compared to the 6-s protocol; (2) the 6-s RD protocol induced larger gains in MVIC at knee flexion of 30° than the 2-s RD, but both protocols induced similar increases in MVIC at for the 90° of knee angle. Thus, RD appears to influence the joint-angle specific maximal isometric strength gains only in shorter muscle lengths.

Both rectus femoris and vastus lateralis muscles had a similar increase in CSA for the 2-s RD and the 6-s RD protocols after 35 training sessions with a trivial effect size ($\eta^2 < 0.010$). Different factors have shown to explain the increase in muscle hypertrophy: elevated EMG amplitude (e.g., higher recruitment of motor units and/or firing rate) (*Hunter, Duchateau & Enoka, 2004*; *Lacerda et al., 2016*), increase in force production (*Sampson & Groeller, 2016*; *Sampson, Donohoe & Groeller, 2014*), and also differences in the training volume and TUT (*Schoenfeld & Grgic, 2018*). In general, a higher training volume is connected to an increase in the TUT. Furthermore, a previous study investigating resistance protocols matched by training volume, more substantial muscle hypertrophy after training was detected with longer RD and TUT (*Tanimoto & Ishii, 2006*). In the present study, the 6-s RD protocol was executed with average TUT approximately 25% larger than in the 2-s RD protocol (43 s vs. 30 s, respectively), while the 2-s RD protocol encompassed double of the 6-s RD training volume (14 vs. 7 repetition per set, respectively). Additionally, the present study also showed higher EMG amplitude for the 2-s compared to 6-s RD protocol, in line with previous findings (*Lacerda et al., 2016*; *Sakamoto & Sinclair, 2012*), which may also have contributed to muscle hypertrophy. Thus, although a larger training volume and higher EMG amplitude were verified in the 2-s RD protocol, the longer TUT in the 6-s RD protocol may have counterbalanced each other and led to similar impact on muscle hypertrophy, explaining the present findings. Therefore, these results suggest that protocols with shorter RD performed with a higher training volume and higher EMG amplitude could lead to muscle hypertrophy, to the same extent as protocols with longer RD performed with longer TUT.

The increase in the number of motor units recruited during resistance training has been pointed out as a central factor to trigger muscle hypertrophy (*Schoenfeld, 2013*). The present study demonstrated a higher EMG amplitude for the 2-s RD as compared to the 6-s RD protocol during most of the ROM measurements conducted (100° to 40°). In contrast, the 6-s RD protocol showed higher EMG amplitude only in the last 10° range for concentric ROM (40° to 30°). This result suggests an increased motor unit recruitment during most concentric ROM for the 2-s RD. However, these differences may not have been sufficient to result in a marked increase in CSA to the 2-s RD protocol as compared to the 6-s RD. Although increased EMG amplitude is associated with higher motor unit recruitment other factors may contribute to the changes in the EMG amplitude as well such as increased firing frequency and synchronization of motor units (*Hunter, Duchateau & Enoka, 2004*). Therefore, care must be taken when interpreting EMG data obtained prior and after the resistance training period.

According to EMG amplitude, higher force values were found during the 2-s RD protocol at the beginning of concentric ROM while higher force values were observed for 6-s RD protocol at the end of concentric ROM. Moreover, at the 39th experimental session, the 2-s

RD protocol was executed with higher muscle forces (100° to 70°) during most ROM areas compared to the 6-s RD protocol (50° to 30°). These results agree with previous results showing higher muscle forces at the beginning of concentric ROM when performing faster movements (*Sampson, Donohoe & Groeller, 2014*). It should be noted, however, that, although the 2-s RD was only a third of the 6-s RD, the higher forces applied in the two experimental situations were similar (43% of MVIC at 39th experimental session). This result does not agree with previous studies (*Sampson, Donohoe & Groeller, 2014*). *Sampson, Donohoe & Groeller (2014)* compared protocols with different RDs and showed differences in force produced close to 20% at the onset of first concentric action. However, their participants were instructed to perform ballistic movements or controlled movements within 4 s. With similar protocols to those in the last study, *Sampson & Groeller (2016)* found similar gains in muscle hypertrophy in both experimental conditions after 12 weeks of training. For these authors, the higher force applied during ballistic movements would be a determinant factor for muscle hypertrophy. Therefore, it is possible that in the present study, the similar force during 2-s RD protocol at the beginning of the concentric ROM and the 6-s RD protocol at the end of concentric ROM was possible the reason behind the lack of difference in the CSA between the different protocols.

In agreement with the CSA overall responses, the average maximal dynamic strength performance (1RM test) was similar between the 2-s RD and the 6-s RD protocols (trivial effect size, $\eta^2 < 0.010$). As a mixed result, higher force values were detected in the 2-s RD as compared to the 6-s RD protocol during the initial phase of the ROM in the concentric action while the 1RM gains were similar between protocols. This outcome did not confirm the trend of superior performance when training with fast movement velocities and moderate intensities (60–79% 1RM), as was indicated in a previous meta-analysis (*Davies et al., 2017*). Hence, our initial hypothesis was rejected.

In line with previous research (*Sampson & Groeller, 2016*), we did not find differences in 1RM performance originating from different RD protocols using the same load. In contrast, other investigators had found higher 1RM gains when utilizing fast movement velocities (*González-Badillo et al., 2014*; *Padulo et al., 2012*). The discrepancy between these results may be associated to different RD adopted in these studies. Although the movement time in the 2-s RD protocol was three times shorter than in the 6-s RD protocol, participants were not instructed to perform explosive movements, which was the case in other studies (*González-Badillo et al., 2014*; *Padulo et al., 2012*). It has been reported a greater EMG amplitude and impulse production when ballistic movements were performed (*Maffiuletti et al., 2016*). This factor may have influenced the occurrence of adaptations favorable to the 1RM increase for faster protocols in these studies (*González-Badillo et al., 2014*; *Padulo et al., 2012*), which was not observed in the present research. Importantly, in the abovementioned studies at least one of the analyzed protocols was not performed to MF (*González-Badillo et al., 2014*; *Sampson & Groeller, 2016*). Thus, differences within the levels of effort must be assumed. Moreover, performing repetitions to MF provide a maximum effort for all individuals during both protocols (*Dankel et al., 2017*), which may have been a determinant factor in not having difference in 1RM gains between the two

protocols investigated in the present study. Therefore, training to MF could hamper the effect of RD on maximum dynamic strength performance observed in previous studies.

In addition, it has been reported that exposure to successive 1RM tests on a three-week basis may bias in the 1RM individual's performance when comparing different resistance protocols (*Morton et al., 2016*). Consecutive measurements may evolve to a similar motor pattern within the 1RM tests. Hence, it appears possible that existing differences between training protocols may not be detected (*Bernardi et al., 1996*). Data from an untrained control limb would enable a better understanding of the effect of repeated exposure of both trained limb to 1RM tests. Thus, the lack of a control limb was a limitation of the present study and must be considered when interpreting the results. Although there are no significant differences in 1RM improvement, the RD effect on maximal strength could be observed when tested with other measurement procedures. (*Pereira & Gomes, 2007*) therefore, MVIC tests performed at different joint angles may be a valid alternative to investigate the effect of different training protocols on muscle strength responses. Given that protocols with different RD provide changes in force production that varies over the ROM (*Sampson, Donohoe & Groeller, 2014*), it was hypothesized that greater isometric maximal strength gains would be found in knee-joint angles with higher instantaneous force values applied.

Similar to the 1RM scores, the increases in the MVIC values at 90° of knee flexion did show comparable improvements by both RD training protocols. A trivial effect size ($\eta^2 < 0.010$) reinforces the result found by ANCOVA. Conversely, larger gains were observed in the MVIC at 30° of knee flexion for 6-s RD protocol. However, a small effect size was observed for the analysis of the interaction effect between protocol and time ($\eta^2 = 0.048$) and must be considered when interpreting the results. Based on results from previous studies (*Alegre et al., 2014*), we hypothesized that the distinct force–angle relationship obtained during the two training protocols would influence the MVIC gains at different knee-joint angles. These previous studies, in general, presented larger maximum strength gains in the joint angles near the training angle/ROM used in the corresponding training. In contrast, in the present study, our subjects performed both training protocols with the same ROM (70°) varying the force generated along with the angular exercise range (100° to 30°). Data from the 39[th] training session (Fig. 8B) show that in the initial (100−90°) and final (40−30°) ranges of the concentric testing action largest differences in force production were detected between the protocols (7 vs. 13%). Additionally, the 2-s RD protocol showed higher values at the beginning and the 6-s RD protocol at the end of the testing movement. Given that type of mechanical stimulus (i.e., type of contraction) may contribute to the specific training adaptations (*Buckthorpe et al., 2015*) it appears possible that the higher force production in the 6-s RD protocol compared to the 2-s RD protocol at the end of concentric action might be sufficient to produce differences in isometric force gains at 30° but not at 90° of knee flexion. Furthermore, increases in isometric force after training have been associated with increases in EMG amplitude (*Noorkoiv, Nosaka & Blazevich, 2014*). The EMG-angle relationship for the 6-s RD protocol showed higher normalized EMG amplitude at 40−30° of knee-joint angle (end of concentric action) and so help to explain the force gains at 30° of knee flexion. Moreover, previous studies verified joint-angle
specific strength gains close to trained angle/ROM (*Alegre et al., 2014*) while others only showed joint-angle specific strength gains for resistance training performed in shorter muscle lengths (*Noorkoiv, Nosaka & Blazevich, 2014*), reinforcing the results obtained in the present study only at 30° of knee flexion. However, it should be emphasized that the mechanisms suggested explaining distinct strength gains at specific joint angles are still poorly understood (*Alegre et al., 2014*; *Noorkoiv, Nosaka & Blazevich, 2014*).

A limitation of the intra-individual experimental design is a possible cross-training or cross-education effect (*Beyer et al., 2016*). There is evidence in the literature indicating that the cross-training effect, if it occurs, would be restricted to neural parameters and muscle strength gains while morphological changes (e.g., CSA) would not be influenced by this effect (*Beyer et al., 2016*). In this respect, muscle strength gains in the contralateral limb should evolve from an increase in the motor neuron activation and are not related to morphological adaptations. However, previous studies investigating the crossing-effect for EMG amplitude showed inconclusive results (*Hortobágyi, Lambert & Hill, 1997*; *Lee & Carroll, 2007*). For example, *Hortobágyi, Lambert & Hill (1997)* found that changes in the EMG amplitude of the untrained limb depending on the training mode performed (e.g., type of muscle action). The neuromuscular changes were similar to the changes in muscle strength. In addition, researchers found that the cross-training effect contributes to approximately 7.8% of the muscle strength gain of the contralateral limb (*Munn, Herbert & Gandevia, 2004*). Such adaptation was explained by neural mechanisms involving acute facilitation within the motor cortex of the untrained contralateral limb following excitation of the trained limb (*Fisher, Blossom & Steele, 2016*). The training protocols were performed with a minimum interval of 24 h in order to minimize the acute effect of unilateral training reducing the maximal strength performance in the contralateral limb. Finally, and most important to our study, it has been argued that, when both limbs of an individual are trained with different protocols, the cross-training effect is minimal or non-existent (*Munn, Herbert & Gandevia, 2004*; *Bell et al., 2020*). Hence, we expected that any difference in the strength responses between limbs would be due to training protocols and not owing to a crossing effect (*Fisher, Blossom & Steele, 2016*).

## CONCLUSIONS

This study showed that protocols with different RD performed to MF produced similar muscle hypertrophy gains despite differences in the EMG amplitude and force–angle relationships. Therefore, different training volumes and TUTs based on the different RDs appear to produce a similar stimulus to skeletal muscle growth. Thus, we argue that an increased training volume provided by performing faster movements to MF would promote similar muscle hypertrophy when compared to higher TUTs during slower movements. It is noteworthy that the highly trained individuals possibly require larger training volumes in order to achieve chronic adaptations (i.e., muscle hypertrophy) associated with resistance training as compared to untrained or moderately trained individuals (*Figueiredo, De Salles & Trajano, 2018*). Yet, although no differences in 1RM gains between protocols were found, our MVIC data provides important insight for the understanding

of joint-angle specific strength responses induced by RDs. We demonstrate that high force production in the end of concentric action during the 6-s RD protocol induced higher maximal isometric strength at 30° of knee flexion when compared to the 2-s RD protocol.

Repetition duration is considered an essential variable of resistance training (*Dankel et al., 2017*; *Davies et al., 2017*; *Pereira & Gomes, 2007*; *Tanimoto & Ishii, 2006*), but recent studies are not supporting this view on strength and muscle hypertrophy (*Davies et al., 2017*; *Schoenfeld, Ogborn & Krieger, 2015*). Nevertheless, this investigation has shown that resistance training performed to MF with longer RD could be a more appropriate strategy to provide greater gains in maximal muscle strength at shortened knee positions, although different RD would induce similar muscle hypertrophy. Thus, the current results have practical applications for individuals seeking health-related improvements in muscular strength and hypertrophy. Overall, it should note that the results presented here are limited to the exercise and subject characteristics similar to those of our current study. However, future studies with females and trained individuals are needed to clarify the impact of protocols with different RDs on the chronic adaptations associated with resistance training performed to MF.

### Funding
This work was supported by the Coordenação de Aperfeiçoamento de Pessoal de Nível Superior (CAPES), the Fundação de Amparo a Pesquisa do Estado de Minas Gerais (FAPEMIG) and the Pró-Reitoria de Pesquisa (PRPq) da Universidade Federal de Minas Gerais. The funders had no role in study design, data collection and analysis, decision to publish, or preparation of the manuscript.

### Grant Disclosures
The following grant information was disclosed by the authors:
Coordenação de Aperfeiçoamento de Pessoal de Nível Superior (CAPES).
Fundação de Amparo a Pesquisa do Estado de Minas Gerais (FAPEMIG).
Pró-Reitoria de Pesquisa (PRPq) da Universidade Federal de Minas Gerais.

### Competing Interests
The authors declare there are no competing interests.

### Author Contributions
- Lucas Túlio Lacerda conceived and designed the experiments, performed the experiments, analyzed the data, prepared figures and/or tables, authored or reviewed drafts of the paper, and approved the final draft.
- Rodrigo Otávio Marra-Lopes performed the experiments, analyzed the data, prepared figures and/or tables, authored or reviewed drafts of the paper, and approved the final draft.
- Marcel Bahia Lanza performed the experiments, prepared figures and/or tables, authored or reviewed drafts of the paper, and approved the final draft.

- Rodrigo César Ribeiro Diniz performed the experiments, analyzed the data, authored or reviewed drafts of the paper, and approved the final draft.
- Fernando Vitor Lima, Hugo Cesar Martins-Costa, Gustavo Ferreira Pedrosa, André Gustavo Pereira Andrade and Armin Kibele performed the experiments, authored or reviewed drafts of the paper, and approved the final draft.
- Mauro Heleno Chagas conceived and designed the experiments, performed the experiments, authored or reviewed drafts of the paper, and approved the final draft.

## Human Ethics

The following information was supplied relating to ethical approvals (i.e., approving body and any reference numbers):

The Federal University of Minas Gerais granted Ethical approval to carry out the study within its facilities (79108117.5.0000.5149).

## Data Availability

Raw data, including cross-sectional area, maximal dynamic strength test, maximal voluntary isometric contraction test, normalized EMG and force-angle relationship data, are available as a Supplemental File.

## Supplemental Information

Supplemental information for this article can be found online at http://dx.doi.org/10.7717/peerj.10909#supplemental-information.

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
