# Peer review of "Resistance training with different repetition duration to failure: effect on hypertrophy, strength and muscle activation"

_PeerJ, doi:10.7717/peerj.10909_

## Round 0.1 · original submission · Major Revisions

Thank you for your manuscript on a very interesting topic. There are several changes and comments that need to be addressed before I can further evaluate the acceptability of your manuscript for publication. Please address all of the reviewers' comments in detail, and highlight the changes you've made or alternately, thoroughly address any areas where you disagree. I look forward to reading the next version of your manuscript.

Reviewer 1 ·

Basic reporting

Some grammatical corrections are provided.

Experimental design

no comment

Validity of the findings

The statistical analyses should be ran using the absolute values as opposed to the percentage change values. Additionally, I would encourage placing less emphasis on the individual responses because the intervention was only ran once, and there is no time matched control group. This makes it very difficult to understand if we are examining true differences or just random error. Although the typical error is used, this only accounts for measurement error and not random biological variability that occurs over time.

Additional comments

The authors examine differences between repetition durations. Some comments are provided below:

Primary Comments:

1)The fact that the values increased in both groups does not provide a good rationale for using percentage changes instead of the raw values. This method will produce differential results depending on the differences in baseline scores across people (i.e. someone with a smaller pre value will have more weight on the analysis). You also run the risk of losing a normal distribution when using percentage changes (note: to test this you would need to check if the differences in percentage changes are normally distributed). It makes more sense to use the raw values here.
https://bmcmedresmethodol.biomedcentral.com/articles/10.1186/1471-2288-1-6

2) Throughout the discussion and conclusion section I would place much less of an emphasis on the individual-level data because you do not have a time matched control group to compare this to. Even though you have the TE reported this is not the TE performed over the same duration as the intervention groups, and thus it is likely to be artificially lower due to less random biological variability that occurs over time.
https://www.ncbi.nlm.nih.gov/pubmed/25823596

3) The discussion on line 487 needs to be changed: “the 2-s RD protocol showed at least larger than or similar effects in muscle hypertrophy than the 6-s protocol”. There were no significant differences for either muscle group tested, so therefore neither condition is better or worse. Unless you run a test of inferiority you cannot make this statement since this statement can’t be made from just looking at the individual level data.

4) Similar to the comment above, in the conclusion the following sentences are not supported by the data which showed no differences in CSA between conditions: “However, when considering individual data, faster movements may result in larger muscle hypertrophy (mainly for vastus lateralis muscle) compared to slower movements. Thus, we argue that an increased training volume provided by performing faster movements to MF would promote greater muscle hypertrophy when compared to higher TUTs during slower movements.”

5)How did you get the estimated sample size of 10? When entering the values into G*Power it provides a sample size of 19. This is using a paired t test; 2 tailed; effect size dz = 0.68; alpha = 0.05; power = 0.8.

6)Why were females excluded when a within subject design was used? Any influence of the menstrual cycle or effect of sex would be negated by the fact that each individuals is only being compared to themselves. A limitation of the study is therefore, that the findings may not be generalizable to females.

7) In the discussion on line 521: “However, despite the importance of increased TUT, especially in the process of muscle hypertrophy, a higher motor unit recruitment may be expected through a protocol with shorter RD and higher volume.” Would you really expect a difference in motor unit recruitment even given that both exercises were taken to failure? Why would one not recruit all motor units if exercising to failure? By the last repetition I would expect the muscle activation to be similar across protocols. In line with this comment, there were differences in EMG noted, but would you expect these differences to still be present when the individual is reaching volitional failure?

Additional Comments:

Line 556 “In contrast, the individual analyses showed a higher proportion of participants with greater hypertrophy after training with the 2-s RDs compared to the 6-s RDs.” This sentence should be deleted since this is likely to occur by random chance anyway.

Line 81: the word “response” is needed after “hypertrophic”

Line 270: the sentence is not complete

Line 450: add “the” before “6th”

Line 475: add “but” after “last”

Line 575 and 576 should be corrected for grammar: “It has been reported a greater neuromuscular activation and impulse production when performing ballistic movements (Maffiuletti et al., 2016).”

Reviewer 2 ·

Basic reporting

see below

Experimental design

see below

Validity of the findings

see below

Additional comments

I thank the authors for the opportunity to review their work. Overall, I think the introduction/results/discussion need to be much more focused on their actual aim. There was a lot of extra discussion that was interesting but had nothing to do with the primary purpose of the study which was to “investigate the effects of two 14-week resistance training protocols on strength, hypertrophy, EMG, and their force relationships.” i.e. the only thing that should be of interest is the comparison between groups with each…for example some of your results/discussion are investigating how this changes with session x protocol x knee joint angle…that is not something that was a purpose of the study as reported in the manuscript. Without a non-exercise control condition, it is difficult to say anything about a change across sessions…more appropriate to discuss is this condition different than this other condition. Overall, I request a much more focused introduction/results/discussion.

The individual response discussion should be removed. The issues with this approach has been discussed recently in two review papers by Atkinson and Dankel. With your results, you would want to take the SEM and multiply it by 2.77 (1.96 x sqrt 2) to get the minimal difference…this would then allow you to be confident who exceeded a change from zero…however, this does not necessarily mean that it is different from another score that did not exceed that threshold. This sort of discussion is misleading and it allows for some to conclude that one intervention was better than another when they were in fact not different.
https://physoc.onlinelibrary.wiley.com/doi/abs/10.1113/EP087712
https://link.springer.com/article/10.1007/s40279-019-01147-0

Specific comments below:

Line 44: superior muscle hypertrophy? There were no significant differences between groups. Probably also worth stating that groups were not different in the strength task that mimicked the way in which they trained (isotonic).


Line 67: how influential is exercise volume for strength? There are a number of studies that show large differences in volume yet similar changes in strength?
https://www.ncbi.nlm.nih.gov/pubmed/28463902
https://www.ncbi.nlm.nih.gov/pubmed/31553889


Line 77: what is meant by “CSA is a well-known estimate of the muscle volume”?


Line 84: I found this paragraph confusing…I was having a difficult time following what the arguments were.

Line 125: testing something that wasn’t trained gives you information about a movement/test that wasn’t performed…gets you around the specificity component…but does that provide information as to adaptations to the exercise they were actually performing?

Line 144: it is not clear to me from the literature or your introduction why you’d expect CSA to be different.

Line 171: your sample size estimation does not appear to be correct. The correct sample size for the numbers you input is n=19. Remember, you are interested in the difference in changes which is what you’d want to power your study on.

Line 227: in the introduction, you discussed potential differences at each site…but then pool them all together…why?

Line 272: this might explain your lack of difference in the 1RM…you are no longer testing your intervention when you do multiple exposures of the test to each group. This could have been accounted for with a non-exercise control…the lack of this group/condition should be noted in the limitations section.

Line 309: what was the sampling rate of your EMG?

Line 323: please report your results in the absolute values..as opposed to the % change.

Line 332: is there a way to run a Levene’s test for paired data? This is specific to between subject designs…did you run this as between subjet?

Line 334: this effect size appears incorrect…you want to the difference between conditions divided by the SD of that difference…you have paired data

Line 340: see earlier point about removing this

Line 368: what are control variables?

Line 423: your EMG-angle relationship and Force-angle relationship paragraphs were difficult to follow.

Line 486: your point #1 is not supported…it was not larger…it wasn’t different

Line 594: what strength response is important? The one you are repeatedly doing or the adaptations to a movement that wasn’t trained?

Line 635: what about when both arms are trained? Is there still a cross-over? Here is a recent paper that might help support your model
https://www.ncbi.nlm.nih.gov/pubmed/31652423
https://www.ncbi.nlm.nih.gov/pubmed/28177712

---

## Round 0.2 · Major Revisions

Thank you for your resubmission. There are a number of issues remaining as highlighted by Reviewer 2. Please attend to these issues and note your changes. As you will read, there are some important suggestions remaining around your statistical methods. I request you address the suggestions thoroughly, and this will likely require a reanalysis. Based upon this reanalysis, as your results may change, this may require substantial reworking of your manuscript. I look forward to reviewing your changes.

Reviewer 2 ·

Basic reporting

See below

Experimental design

See below

Validity of the findings

see below

Additional comments

I thank the authors for the opportunity to review the manuscript again and I appreciate the thoughtful responses to my previous points. However, I still have a number of points of consideration…particularly as it relates to your results and interpretation.

1) You state the importance of repetition duration but I would be very cautious with this interpretation because you found that benefit in only one of your variables (30 degrees). It might be worth stating this as more of an invitation for future work to investigate this, particularly since this seemed counter to your initial hypotheses.

2) The G-power output you provide is interesting because it is counter to what you get with known calculations of the t-distribution. For example, you are interested in the difference in differences which is the interaction….this is equivalent to a paired t-test on the change scores, yet G-power seemingly treats it differently (square the F value gives you the t in this scenario).

3) I have the following points with the statistical analysis
A) there is no need to test baseline differences, the limbs were randomly assigned to a treatment and any difference at baseline would be meaningless for interpretation. The purpose of randomization is random assignment not perfect numerical equality.
https://www.ncbi.nlm.nih.gov/pmc/articles/PMC4310023/

B) how did you run a repeated measures ANCOVA where all factors are within? I don’t think this is necessary based on point A but I am not sure how it was completed in SPSS?

C) the Mann-Whitney Wilcoxin test is a between subject test whereas you have within subject data. An assumption of this test is that here are two independent groups…however your groups have dependency (same person).

4) In the discussion section (line 482) you state that repetition duration is an important variable to be associated with chronic adaptations. This comes after discussing the force-angle relationships with EMG…however, is there evidence that you have differences between conditions across time? Based on your figures it seems like the differences present at beginning of training period are the same differences present at the end of training? In other words, there wasn’t really an adaptation per se? From what I can tell, the only value that changed differently between conditions was MVC torque at 30 degrees?

Specific comments found below:

5) Line 67: can you rewrite this sentence for clarity.

6) Line 83: a more accurate depiction…compared to what?

7) in the results section on line 379: should this p value be 0.377?

8) it might be better to refer to muscle activation as “EMG amplitude” since that is what was measured in your study and the Schoenfeld paper you reference.

9) Line 517-535: since this doesn’t appear to differ between protocols, I am not sure it needs a full paragraph of discussion.

10) line 563: please double check Morton paper…I believe it may have been every 3 weeks.

11) in your figures, please connect the individual dots so we know which one came from which person. In other words, connect their Post to their Pre…otherwise the individual plots are not all that informative.

---

## Round 0.3 · Minor Revisions

Thank you for your much improved manuscript. I have received Reviewer #2's report who suggests some small, but important, changes. Please attend to these changes. Thank you,
Scotty

Reviewer 2 ·

Basic reporting

The individual plots for the figures are still not correct. You connect the pre test values of one limb with the pre test value of the opposite limb. This is not informative for describing the “response” to the intervention. The pre value for one condition needs to be connected with the post test value of the same condition. This needs to be corrected throughout.

Experimental design

n/a

Validity of the findings

n/a

Additional comments

n/a

---

## Round 0.4 · accepted · Accept

Thank you for your attention to the reviewers' comments. This manuscript is now acceptable for publication. Congrats! Scotty